# Confidence-based Reliable Learning under Dual Noises

**Peng Cui[1 3], Yang Yue[1], Zhijie Deng[1 2]\*, Jun Zhu[1]\***
[1] Dept. of Comp. Sci. & Tech., Institute for AI, BNRist Center,
Tsinghua-Bosch Joint ML Center, THBI Lab, Tsinghua University, Beijing, 100084 China
[2] Qing Yuan Research Institute, Shanghai Jiao Tong University      [3] RealAI
`xpeng.cui@gmail.com, yueyang22@mails.tsinghua.edu.cn`
`zhijied@sjtu.edu.cn, dcszj@tsinghua.edu.cn`

## Abstract

Deep neural networks (DNNs) have achieved remarkable success in a variety of computer vision tasks, where massive labeled images are routinely required for model optimization. Yet, the data collected from the open world are unavoidably polluted by noise, which may significantly undermine the efficacy of the learned models. Various attempts have been made to reliably train DNNs under data noise, but they separately account for either the noise existing in the labels or that existing in the images. A naive combination of the two lines of works would suffer from the limitations in both sides, and miss the opportunities to handle the two kinds of noise in parallel. This works provides a first, unified framework for reliable learning under the joint (image, label)-noise. Technically, we develop a confidence-based sample filter to progressively filter out noisy data without the need of pre-specifying noise ratio. Then, we penalize the model uncertainty of the detected noisy data instead of letting the model continue over-fitting the misleading information in them. Experimental results on various challenging synthetic and real-world noisy datasets verify that the proposed method can outperform competing baselines in the aspect of classification performance.

## 1   Introduction

Deep Neural Networks (DNNs) have obtained great success in a wide spectrum of computer vision applications [26, 41, 19, 18], especially when a large volume of carefully-annotated low-distortion images are available. However, the images collected from the wild in real-world tasks are unavoidably polluted by noise in the images themselves (e.g., image corruptions [20] and background noise [43]) or the associated labels [36], termed as image noise (*x-noise*) and label noise (*y-noise*) respectively. Previous investigations show that the DNNs naively trained under *y-noise* [2, 52] or *x-noise* [11, 53] suffer from detrimental over-fitting issues, thus exhibit poor generalization performance and serious over-confidence.

There has been a large body of attempts towards dealing with data noise, but they mainly focus on a limited setting, where noise only exists in either the label (i.e., noisy labels) [36, 1, 31, 8] or the image [13, 27, 50]. It is non-trivial to extend them to exhaustively deal with dual noises (i.e., the joint *(x,y)-noise*). Moreover, the techniques for handling *x-noise* suffer from non-trivial limitations. For example, most image denoising methods work on well-preserved image texture [12], thus may easily fail when facing images that are globally blurred (see Fig. 5 in Appendix); alternative image Super-Resolution (SR) solutions are usually computationally expensive [46]. These issues raise the requirement of a unified approach for reliable learning under dual noises.

---

[*]The corresponding author.

Compared to deterministic DNNs, uncertainty-based deep models (e.g., Bayesian Neural Networks (BNNs) [3] and *deep ensemble* [25]) reason about the uncertainty and hence have the potential to mitigate the over-fitting to noisy data. Empowered by this insight, we first perform a systematical investigation on leveraging uncertainty-based deep models to cope with dual noises. We observe that, despite with less over-fitting, the uncertainty-based deep models may still suffer from the bias in the noisy data and yield compromising results.

To further ameliorate the pathologies induced by data noise and achieve reliable learning, we propose a novel workflow for the learning of uncertainty-based deep models under dual noises. Firstly, inspired by the recent success of using predictive confidence to detect the out-of-distribution data [21], we propose to detect both the noisy images and the noisy labels by the predictive confidence produced by uncertainty-based deep models. Concretely, we use the predictive probability corresponding to the label (i.e., label confidence) to filter out the samples with *y-noise*, and use the maximum confidence to filter out the samples with *x-noise*. After doing so, we propose to penalize the uncertainty [23] of the detected noisy data to make use of the valuable information inside the images without relying on the misleading supervisory information.

Given the merits of *deep ensemble* [25] for providing calibrated confidence and uncertainty under distribution shift revealed by related works [37] and our studies, we opt to place our workflow on deep ensemble to establish a strong, scalable, and easy-to-implement baseline for learning under dual noises. Of note that the developed strategies are readily applicable to other uncertainty-based deep models like BNNs.

We perform extensive empirical studies to evidence the effectiveness of the proposed method. We first show that the proposed method significantly outperforms competitive baselines on CIFAR-100 and TinyImageNet datasets with different levels of synthetic *(x,y)-noise*. We then verify the superiority of the proposed method on the challenging WebVision benchmark [28] which contains extensive real-world noise. We further provide insightful ablation studies to show the robustness of our approach to multiple hyper-parameters.

## 2 Related Work

Many methods have been proposed to deal with *y-noise* in deep learning. A direct approach is to design the robust loss functions, e.g., the loss function based on the mean absolute error [16] and the symmetric cross-entropy [45, 6]. However, it is challenging to deal with the noisy data with high noise rates. An alternative method is to train on reweighing or selected training examples, e.g., estimating the weight of samples based on meta-learning [17], MentorNet [22] and Co-teaching [40], but designing an effective algorithm or criterion of selecting the samples based on the deterministic DNNs tends to be difficult. Recently, the loss correction approaches are also used to mitigate the over-fitting to noisy labels by assigning a weight to the prediction of the model [39, 1] or by adding a regularization to the loss function [31, 8]. To deal with *x-noise*, image denoising may be a useful technique. [13] assumes a uniform camera blur over the image and then applies a standard deconvolution algorithm to reconstruct the blurry image, but it can only handle those locally-blurred images. [50, 34] propose to use a deep convolutional neural network to capture the characteristics of degradation and restorate blurred images, but they commonly need image pairs (i.e., the label indicates clean or noisy) for training and the supervised information cannot be provided in our setting. Therefore, it is not free to extend the existing works to handle dual noises, and developing new techniques is necessary.

Typically, in machine learning and computer vision, the uncertainty we are concerned about can be classified into two categories: *Epistemic* uncertainty and *Aleatoric* uncertainty, which are also called model uncertainty and data uncertainty [23]. Extensive uncertainty quantification approaches have been proposed in the literature. A direct approach to incorporating uncertainty into DNNs is to perform Bayesian inference over DNN weights, with BNNs [3, 30], Monte Carlo (MC) dropout [15] and SWAG [32] as popular examples. Yet, Bayesian inference is often expensive due to the high non-linearity of DNNs. An alternative way is to adapt various distance-aware output layers into DNNs in a non-Bayesian way [44, 29, 33]. However, these methods may suffer from degenerated uncertainty estimates [14] due to restrictive assumptions. *Deep ensemble* [25] is a prevalent and leading tool for uncertainty quantification, which can produce calibrated confidence and uncertainty [37] by

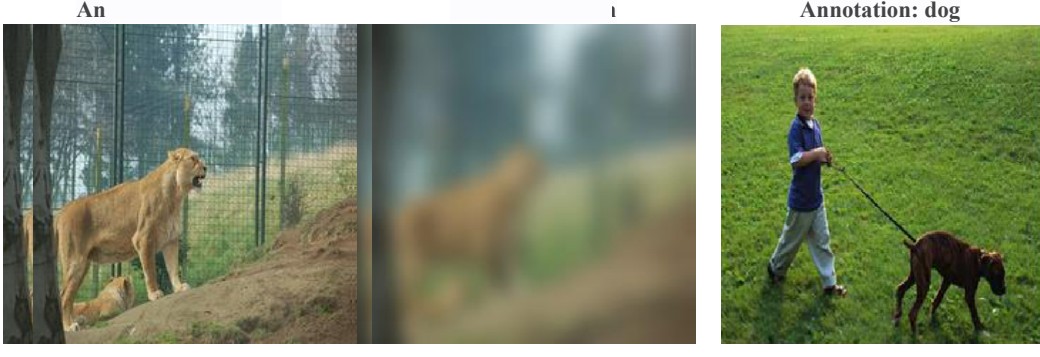

| (a) *y-noise*: wrong annotations. | (b) *x-noise* I: corrupted images. | (c) *x-noise* II: background noise. |

Figure 1: An illustration of *y-noise* and *x-noise*.

assembling the outputs of different DNN predictors uniformly. In this paper, we place our workflow on deep ensemble to establish a robust learning approach under dual noises.

## 3 Preliminaries and Problem Setting

Let $\mathcal{D} = \{(x_i, y_i)\}_{i=1}^{N}$ denote a collection of image-label pairs, with $x_i \in \mathbb{R}^d$ and $y_i \in \{1, 2, ..., C\}$ as the image and the label respectively. We can routinely deploy a $\theta$-parameterized classifier (e.g., a DNN) $f_\theta : \mathbb{R}^d \to \Delta^C$ for data fitting, where $\Delta^C$ is the probability simplex over $C$ classes. In other words, the classifier defines a probability distribution $p_\theta(y|x) = p(y|f_\theta(x))$. Typically, we minimize the cross-entropy loss, i.e., perform maximum likelihood estimation (MLE), to train the model:

$$\min_\theta \ell(\theta; \mathcal{D}) = -\frac{1}{N} \sum_{i=1}^{N} \log\left(f_\theta\left(x_i\right)\left[y_i\right]\right), \tag{1}$$

where $f_\theta(x_i)[y_i]$ refers to the $y_i$-th element of the vector $f_\theta(x_i)$. We can also augment the above objective with an L2 penalty on weights $||\theta||_2^2$ to achieve maximum a posteriori (MAP) estimation.

To enable the characterization of uncertainty, Bayesian neural networks (BNNs) place a prior distribution over DNNs weights $p(\theta)$, and perform Bayesian inference to find the posterior distribution $p(\theta|\mathcal{D})$ instead of performing MLE or MAP estimation as in the deterministic DNNs. Such an uncertainty-aware modeling can give rise to a more calibrated predictive distribution.

### 3.1 The Setting of Learning under Noise

In practice, the collected dataset may suffer from heterogeneous *noise*. A typical assumption on data noise is that there are systematical errors in the annotations, i.e., the label noise (*y-noise*). For example, an image of lion may be annotated as "tiger" as shown in Fig. 1a. Tremendous effort has been devoted to handling symmetric, asymmetric, or even instance-dependent *y-noise* [38, 45, 49]. However, in practice, the noise may exist in not only the annotations but also the images themselves (i.e., *x-noise*), casting new challenges for the deep learning models in the real world.

Common *x-noise* includes image corruptions [20] like distortion, blur, compression, etc. (see Fig. 1b). The *x-noise* may also stem from the inherent ambiguity of the image (see Fig. 1c), which is termed as background noise by the previous work [43]. We use *x-noise* I and *x-noise* II to refer to the aforementioned two types of *x-noise* for short. The *x-noise* results in low-quality or even incomplete observations and may cause over-fitting and bias the model. The existing works for dealing with *x-noise* mainly focus on image corruptions (e.g., image denoising and Super-Resolution (SR)), and often require some specific assumptions [12] or expensive computational resources [46]. Namely, there are still barriers for them to deal with the real-world image noise [12], especially for the background noise.

In this paper, we focus on the learning under dual noises (i.e., the joint *(x,y)-noise*), a more general and more challenging setting than learning under only *x-noise* [20, 43] or *y-noise* [38, 49]. A naive

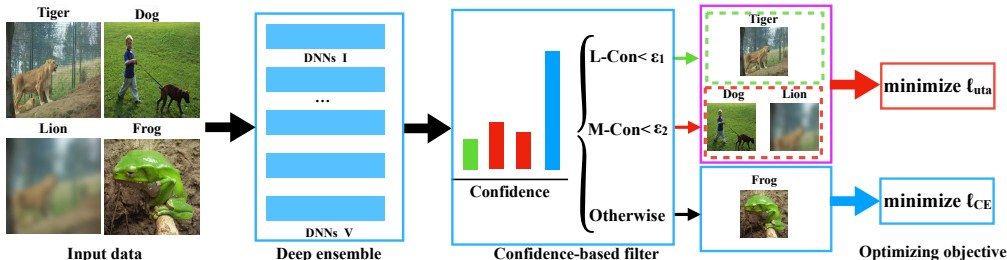

Figure 2: Overview of the proposed method. Given training data with *(x,y)-noise*, the proposed method first distinguishes noisy samples from clean samples using the *confidence-based sample filter*. Then, we can minimize the standard cross-entropy loss for clean data but minimize model uncertainty for noisy data in the framework. $\ell_{uta}$ represents the loss function of uncertainty penalty.

combination of the two lines of works would suffer from the limitations in both sides, and miss the opportunities to handle x-noise and y-noise in parallel. To address this challenge, we need to develop a unified and reliable learning strategy to avoid detrimental over-fitting.

## 4 Methodology

Uncertainty-based deep models can potentially mitigate the over-fitting to noisy data due to the inherent characterization of uncertainty. We have conducted a thorough empirical study on using uncertainty-based deep models like BNNs and deep ensemble to handle dual noises (see Appendix A). We found that uncertainty-based models can better alleviate over-fitting than deterministic DNNs. However, these models can still suffer from the bias in the noisy data and yield compromising results. As a result, we propose two strategies to further promote the effectiveness of uncertainty-based deep models for handling dual noises. We place our following discussion upon deep ensemble, one of the best uncertain-based deep models revealed by pioneering works [37] and our study, and clarify that our strategies are compatible with other backbones like BNNs.

We first briefly review deep ensemble. Concretely, a deep ensemble consists of $M$ randomly initialized, individually trained DNNs $\{f_{\theta_m}\}_{m=1}^M$, and makes predictions by uniform voting:

$$\frac{1}{M}\sum_{m=1}^M f_{\theta_m}(x). \tag{2}$$

As shown, deep ensemble is easy to implement and flexible, which makes our approach enjoys good practicability and scalability.

In the following, we discuss how to construct a confidence-based sample filter to progressively filter out noisy samples, and how to excavate valuable information from detected noisy data. We illustrate our method in Fig. 2.

### 4.1 The Confidence-based Sample Filter

Distinct from leveraging complicated strategies for noise detection in previous works [1, 31], we propose a simple confidence-based sample filter to filter out *x-noise* and *y-noise* in parallel.

**Filtering out *y-noise* using the Label confidence (L-Con).** Specifically, we first use the predictive probability corresponding to the label $y$ (i.e., the label confidence) to distinguish the data with *y-noise* from the others. In the case of deep ensemble, the label confidence can be simply estimated by:

$$\textbf{L-Con}(x) = \frac{1}{M}\sum_{m=1}^M f_{\theta_m}(x)[y]. \tag{3}$$

Intuitively, **L-Con** reflects how confident the model is for the current input w.r.t. the label. Our hypothesis is that our model tends to yield low **L-Con** for the training data with *y-noise* yet yield high **L-Con** for the others. We empirically corroborate this in Fig. 4 in Appendix. As shown, the data

with *y-noise* can be accurately distinguished from the clean data by **L-Con**. More importantly, the **L-Con** of the data with *y-noise* is not mixed up with that of the clean data even at the later training phase (see Fig. 4b).

**Filtering out *x-noise* using the Maximum confidence (M-Con).** We then move to the detection of *x-noise*. Inspired by the success of using the maximum confidence for out-of-distribution detection [21], we utilize the maximum confidence to detect the training data with *x-noise*. The maximum confidence of deep ensemble takes the form of

$$\textbf{M-Con}(x) = \max_j \left( \frac{1}{M} \sum_{m=1}^{M} f_{\theta_m}(x) \right)[j]. \tag{4}$$

**Why M-Con is effective in detecting *x-noise*?** In fact, there is an inherent connection between **M-Con** and data uncertainty (i.e., aleatoric uncertainty). Recalling the explanation in [23, 10], the data/aleatoric uncertainty represents the magnitude of the inherent data noise (e.g., sensor noise), and can be estimated by

$$\text{Ale}(x) = \mathbb{E}_{p(\theta|\mathcal{D})}(\mathcal{H}(p(y|x,\theta))) = \frac{1}{M} \sum_{m=1}^{M} \mathcal{H}(p_{\theta_m}(y|x,\theta)),$$

where $\mathcal{H}$ is the Shannon entropy, and it can be directly estimated by

$$H[p(y|x)] = -\sum_{c=1}^{C} (f_\theta(x)[c])(\log f_\theta(x)[c]), \tag{5}$$

where $C$ is the number of classes. When the model is confident in its prediction (i.e., **M-Con** is high), it yields a sharp predictive distribution centered on one of the corners of the simplex. In contrast, when the model is not confident in its prediction (i.e., **M-Con** is low), it yields a flat predictive distribution scattered in every direction of the simplex, which corresponds to a high data uncertainty. There is evidence showing that the data uncertainty grows as the quality of the input image degrades [5], so **M-Con** is effective in detecting the noisy data with *x-noise*.

Besides, **M-Con** is effective to detect the data with underlying complexity or bias. As shown in Fig. 7 in Appendix, some samples with low **M-Con** correspond to the data with underlying complexity or bias (i.e., hard or dirty samples). As evidenced by some closely related works [35, 5], detecting the data with underlying complexity (bias) for a particular treatment can improve the model's predictive performance or training efficiency.

**How to filter?** We propose a simple yet efficient sample filter based on **L-Con** and **M-Con**. To be specific, we first assign different weights for different data according to the value of **L-Con**,

$$w_i^l = \begin{cases} 0, & \text{if } \textbf{L-Con}(x_i) \leq \epsilon_1 \\ 1, & \text{otherwise,} \end{cases} \tag{6}$$

where $\epsilon_1$ is the threshold for filtering out *y-noise*, and $w_i^l$ indicates whether the label of input sample is noisy ($w_i^l = 0$) or clean ($w_i^l = 1$). Likewise, we can also filter out the samples with *x-noise* according to the value of **M-Con**:

$$w_i^k = \begin{cases} 0, & \text{if } \textbf{M-Con}(x_i) \leq \epsilon_2 \\ 1, & \text{otherwise.} \end{cases} \tag{7}$$

$\epsilon_2$ is the threshold to decide whether the input sample is clean ($w_i^k = 1$) or not ($w_i^k = 0$).

After twice filtering, the final sample weight is $w_i^s = w_i^l \times w_i^k$. Generally, we first train the deep ensemble under a high learning rate for some epochs, after which we use the confidence-based sample filter to filter out noisy data at per iteration. The foregoing warm-up can make the sample filter better for distinguish the noisy data from the clean one.

Furthermore, we perform quantitative experiments to demonstrate the efficacy of the filters for detecting noisy data with *x-noise* and *y-noise*. Concretely, we regard the detection of noisy data as a binary classification problem and use the Area Under the Receiver Operating Characteristic curve (AUROC) to indicate the effectiveness of our filter. As shown in Table 6 in Appendix, the confidence-based (i.e., **M-Con** and **L-Con**) sample filter can achieve high AUROCs.

## 4.2 Uncertainty Penalty on Noisy Data

We first discuss the limitations of the typical learning objectives for dealing with dual noises. Then, we propose an improved learning objective based on model uncertainty.

**Limitations of typical learning objectives.** After distinguishing the clean samples from the noisy ones, it is necessary to resort to some new learning objectives to drive the model training, since that continuing pushing the model to fit dual noises may exacerbate the over-fitting. Typical strategies like the loss correction technique [39, 1] regard the model predictions as pseudo labels and minimize the following loss

$$\ell(\theta; \mathcal{D}) = -\sum_{i=1}^{N} \left( \alpha_i \log \left( f_\theta \left( x_i \right) [y_i] \right) + \beta_i \sum_{c=1}^{C} \texttt{stop\_grad}(f_\theta(x_i)[c]) \log(f_\theta(x_i)[c]) \right), \quad (8)$$

where $\alpha$ and $\beta$ are the weights for clean data and noisy labels.

Nevertheless, it is non-trivial to extent these strategies to dealing with the data with *x-noise*. The model cannot make reliable predictions for the images with *x-noise*, so taking them as pseudo labels may be harmful.

**The model uncertainty estimation.** Fortunately, we notice that deep ensemble can offer high-quality measures of model uncertainty for the input data [25, 37]. By penalizing the model uncertainty of noisy data, we can make our model certain on the training data with *(x,y)-noise*. Specifically, the model uncertainty can be measured by the mutual information between the predictions and the model parameters [10, 42].

$$\underbrace{\mathcal{I}\left[y, \theta | x; \mathcal{D}\right]}_{\text{Model Uncertainty}} = \underbrace{\mathcal{H}\left[ \mathbb{E}_{P(\theta|\mathcal{D})}\left(p(y|x, \theta)\right) \right]}_{\text{Total Uncertainty}} - \underbrace{\mathbb{E}_{P(\theta|\mathcal{D})}[\mathcal{H}(p(y|x, \theta))]}_{\text{Data Uncertainty}},$$

which, in the case of deep ensemble, boils down to

$$\mathcal{I}\left[y, \theta | x; \mathcal{D}\right] \approx \mathcal{H}[\frac{1}{M} \sum_{m=1}^{M} p_{\theta_m}(y|x)] - \frac{1}{M} \sum_{m=1}^{M} \mathcal{H}[p_{\theta_m}(y|x)]. \quad (9)$$

**The proposed learning objective.** Specifically, we optimize the following loss for each ensemble member in deep ensemble:

$$\min_{\theta_m} \ell(\theta_m; \mathcal{D}) = \begin{cases} \sum_{i=1}^{N} -\log\left(f_{\theta_m}\left(x_i\right)[y_i]\right), & \text{if } w_i^s = 1 \\ \sum_{i=1}^{N} \mathcal{I}(y, \theta | x_i, \mathcal{D}), & \text{if } w_i^s = 0 \end{cases} \quad (10)$$

where $w_i^s$ is the weight of each sample. Namely, we minimize the standard cross-entropy loss for clean data, but minimize the model uncertainty for noisy data. Intuitively, the former allows the model to constantly learn useful information when the labels and images are reliable. The latter enables the model to explore the valuable information inside the noisy data, while preventing the model from being misled by the harmful supervisory information. We detail the whole process of the proposed method in Algorithm 1.

## 5 Experiment

In this section, we first evaluate the proposed method on datasets with synthetic noise and the real-world dataset WebVision. Furthermore, we ablate the robustness of the proposed method to hyper-parameters in terms of the number of ensembles: $M$ and two thresholds: $\epsilon_1$ and $\epsilon_2$. We also verify the effectiveness of the uncertainty penalty strategy in ablation studies.

**Datasets.** The proposed method is first evaluated on two benchmark datasets with synthetic noise: CIFAR-100 [24] and TinyImageNet [24] (the subset of ImageNet[9]), the former consists of 100 classes with 32x32 color images, and the latter has 200 classes with 64x64 color images. Moreover, we validate the effectiveness of the proposed method under more challenging real-world noise on WebVision [28], which contains more than 2.4 million images crawled from the Flickr website and Google Images search.

---

**Algorithm 1:** Training DNNs under *(x,y)-noise*

---

**Input:** Training noisy dataset $\mathcal{D}$, number of networks $M$ for ensemble, **L-Con** threshold $\epsilon_1$, **M-Con** threshold $\epsilon_2$

1   Initialize $M$ networks $f_{\theta_1}, \cdots, f_{\theta_M}$;
2   **for** $m = 1 : M$ **do**
3     $\theta^{(m)} \leftarrow \text{WarmUp}(\mathcal{D}, \theta^{(m)})$;
4   **end**
5   **while** $e < \text{MaxEpoch}$ **do**
6     **for** *Mini-batch $\mathcal{B}$ in $\mathcal{D}$* **do**
7       Compute **L-Con** and **M-Con** using equation 3 and 4;
8       Determine weights $w_i^l$ and $w_i^k$ following thresholding rule 6 and 7;
9       Update each network $f_{\theta_m}$ with loss function
        $\mathcal{L}(\theta_m, \mathcal{B}) = \sum_{(x_i, y_i) \in \mathcal{B}} (1 - w_i^k w_i^l) \mathcal{I}(y_i, \theta) + w_i^k w_i^l \mathcal{L}_{\text{CE}}(\theta_m, \mathcal{B})$;
10     **end**
11     $e = e + 1$
12   **end**

---

**Implementation details.** The synthetic noise contains the common *y-noise* used in [51, 1] and *x-noise* I: the corruption on images. We use the symmetric noise as the synthetic *y-noise*, which is generated by randomly flipping the true label to other possible labels. For *x-noise* I, we randomly apply the challenging "Gaussian Blur", "Fog" and "Contrast" corruption used in [20] to the original images to simulate the real-world image noise. The *x-noise* II (i.e., background noise) commonly exists in web images, thus we also evaluate the proposed method on WebVision dataset. The deep ensemble we used consists of 5 ResNet18 [19] for all datasets. SGD is used to optimize the network with a batch size of 256. More details can be found in Appendix B.

**Baselines.** The first thing to note is that all methods employ 5 networks for fair comparisons. We compare with two kinds of compared baselines. The first kind contains the single model (Single-CE) and deep ensemble (DE-CE) with the standard cross-entropy loss. The second kind is competitive loss correction technique related to our method, which involves the regularized loss function with dynamic bootstrapping (DYR) [1], the regularized loss function with mixup dynamic bootstrapping (M-DYR) [1] and COnfidence REgularized Sample Sieve (CORES$^2$) [8]. Besides, we use "Proposed-L (Proposed-M)" to indicate that we only use **L-Con** (**M-Con**) to filter out noisy samples and use "Proposed-LM" to represent the proposed method with **L-Con** and **M-Con** filter. Furthermore, we also consider the pipeline of combining the denoising technique and M-DYR as a compared baseline. However, as shown in Fig. 5 in Appendix, we can observe that existing denoising methods do not restore globally blurred images. As a consequence, a more effective strategy is to filter out low-quality images in this paper rather than restore them.

### 5.1 Performance under Synthetic (x,y)-noise

In this section, we first empirically evaluate the proposed method and other baselines on CIFAR-100 and TinyImageNet with different levels of synthetic *(x,y)-noise*. Afterward, we also compare the proposed method with competitive baselines under the label noise.

We evaluate the classification accuracy at the best and last epoch following the setting of [1]. Table 1 presents the results of all methods on CIFAR-100 and TinyImageNet with different rates of *x-noise* and *y-noise*. We can see that the proposed method outperforms other baselines under synthetic *(x,y)-noise* in terms of classification accuracy at the best and last epoch. Especially, the proposed method achieves a remarkable performance improvement comparing other methods under the joint *(x,y)-noise* (i.e., "$0.2y+0.3x$" and "$0.4y+0.3x$" in Table 1), which shows the effectiveness of the proposed method to handle dual noises.

Besides, we can observe that Proposed-M outperforms DE-CE under *x-noise* (i.e, "$0.3x$" and "$0.4x$"), which shows that the effectiveness of employing **M-Con** to filter out samples with *x-noise*. By contrast, the previous works that focus on the noisy label (i.e., DYR, M-DYR and CORES$^2$) do not show the superior performance regardless of whether *x-noise* or *(x,y)-noise*, which confirms that they cannot effectively handle dual noises. Moreover, we can notice that the naive deep ensemble with

Table 1: The comparison of validation accuracy on CIFAR-100 and TinyImageNet with *(x,y)-noise*. "$0.2y + 0.3x$" represents the dataset with 20% *y-noise* and 30% *x-noise* simultaneously.

| Alg./Noise rate | | 0.0 | 0.3x | 0.4x | 0.2y+0.3x | 0.4y+0.3x |
|---|---|---|---|---|---|---|
| | | CIFAR-100 / TinyImageNet | | | | |
| Single-CE | Best | 77.23/61.19 | 73.62/54.39 | 72.53/52.53 | 57.84/43.59 | 47.76/40.62 |
| | Last | 76.44/60.26 | 72.19/49.03 | 71.95/49.95 | 57.39/36.81 | 41.39/22.62 |
| DE-CE | Best | 79.13/63.62 | 77.07/60.03 | 76.12/59.94 | 66.50/50.03 | 54.90/46.36 |
| | Last | 77.01/61.28 | 76.14/59.51 | 74.98/59.05 | 65.24/46.21 | 53.91/41.27 |
| DYR [1] | Best | 78.64/**65.14** | 73.64/60.74 | 71.68/59.20 | 62.54/52.14 | 50.54/43.94 |
| | Last | 78.02/63.97 | 73.07/59.25 | 71.13/58.01 | 60.59/50.67 | 49.21/40.89 |
| M-DYR [1] | Best | 75.38/62.32 | 75.11/60.70 | 73.86/59.45 | 72.38/52.14 | 64.07/50.50 |
| | Last | 74.91/61.04 | 74.28/58.44 | 72.41/57.21 | 70.69/50.04 | 62.34/48.02 |
| CORES$^2$ [8] | Best | 76.76/59.74 | 73.15/57.22 | 72.04/55.67 | 63.06/46.40 | 51.98/44.55 |
| | Last | 76.22/59.14 | 73.01/56.35 | 71.98/54.41 | 62.51/44.91 | 51.11/43.20 |
| Proposed-L | Best | 80.44/64.07 | 77.01/60.68 | 76.06/59.54 | 71.05/57.62 | 63.04/51.41 |
| | Last | 79.03/63.21 | 76.58/59.99 | 75.08/58.62 | 69.91/56.31 | 61.97/50.23 |
| Proposed-M | Best | 80.61/64.37 | 77.89/**61.06** | 77.51/**60.51** | -/- | -/- |
| | Last | 79.39/64.01 | 77.02/**60.26** | 77.19/**59.34** | -/- | -/- |
| Proposed-LM | Best | **80.98**/64.58 | **77.92**/61.01 | **77.53**/60.12 | **72.78/58.75** | **66.61/52.35** |
| | Last | **79.71/64.15** | **77.03**/60.23 | **77.32**/59.19 | **72.48/57.82** | **66.05/51.02** |

cross-entropy loss (DE-CE) significantly outperforms the single model (Single-CE), confirming that uncertainty-based deep ensemble can prevent the model from over-fitting noisy data. In addition, we can notice that the experimental results exhibit quite close best accuracy and last accuracy, which shows that our method is not easy to over-fit noisy data and can achieve stable and robust learning.

To verify the effectiveness of the proposed method under label noise (i.e., *y-noise*), we also compare our method with other baselines on CIFAR-100 and TinyImageNet with different levels of synthetic *y-noise* in Appendix C. The experimental results show that the proposed method significantly outperforms competitive methods for noisy labels. Specifically, "Proposed-L" also outperforms or is close to the best results of other baselines. We discuss more details in Appendix C.

Table 2: The comparison of validation accuracy on ImageNet ILSVRC12 and WebVision validation set. The number outside (inside) the parentheses denotes top-1 (top-5) accuracy.

| Val./Methods | | DYR | M-DYR | CORES$^2$ | DE-CE | Proposed-LM |
|---|---|---|---|---|---|---|
| WebVision | Best | 69.48 (83.21) | 72.36 (87.40) | 70.56 (87.56) | 73.76 (88.13) | **76.68 (91.32)** |
| | Last | 68.53 (82.42) | 72.01 (87.15) | 69.52 (87.02) | 73.22 (87.98) | **76.52 (91.22)** |
| ILSVRC12 | Best | 67.32 (89.76) | 68.52 (86.36) | 64.12 (86.36) | 67.64 (88.73) | **71.40 (90.88)** |
| | Last | 66.59 (88.98) | 68.33 (86.21) | 63.23 (85.44) | 67.31 (88.26) | **71.26 (90.70)** |

## 5.2 Performance on the Real-world Noisy Dataset

Furthermore, we verify the generalization performance of the proposed method on a large real-world noisy dataset: WebVision. Since the dataset is too big, for quick experiments, we compare all methods on the first 50 classes (denoted as WebVision-50) of the Google image subset and use the resized images following previous works [22, 7]. Besides, we test the trained model of all methods on the human-annotated WebVision validation set and the ILSVRC12 validation set [9]. Table 2 lists the experimental results. As we can see, the proposed method significantly outperforms other baselines not only on the WebVision validation set but also on the ILSVRC12 validation set for the real-world noisy dataset, which shows the superiority of our method is also effective to the real-world noisy dataset.

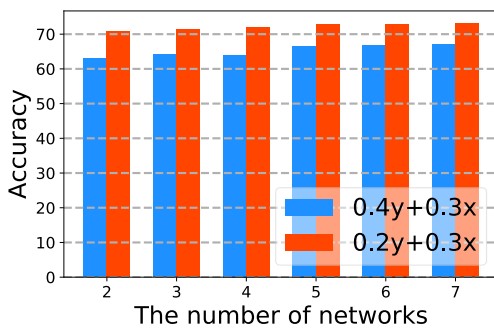 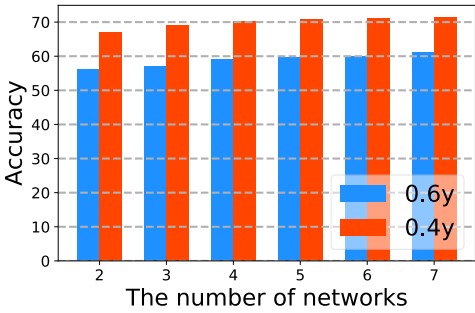

(a) The performance under different levels of (x,y)-noise.  (b) The performance under different levels of y-noise.

Figure 3: Effects of different numbers of networks on the performance of the proposed method on CIFAR-100.

## 5.3 Ablation Studies

**Empirical effects of the number of networks $M$.** The number of networks for deep ensemble is a crucial hyper-parameter. Empirically, the more networks for deep ensemble, the more powerful performance can achieve. However, assembling a large number of networks often requires high memory and computational costs. Hence, we need to make an appropriate trade-off between the performance and the computational cost. Fig. 3 demonstrates the performance (i.e., the best validation accuracy) of the proposed method corresponding to the different numbers of networks under different levels of *(x,y)-noise* on CIFAR-100. We can see that even a small number of networks can not overly drop the performance. When the number of networks is greater than 4, the proposed method can almost achieve the best performance, so an ensemble of 5 networks is enough for our method.

**Empirical effects of thresholds of confidence-based sample filter.** Moreover, we analyze the effects of hyper-parameters: $\epsilon_1$ and $\epsilon_2$ of the proposed confidence-based sample filter on the predictive performance. For the threshold of **M-Con**, we use a soft threshold to filter out the training data with *x-noise* after per iteration (i.e., the training data with minimum $\epsilon_2\%$ **M-Con** is filtered out), which is more effective than the hard threshold through empirical studies. For the threshold of **L-Con**, the hard threshold is more appropriate according to the empirical results in Fig. 4. Table 3 reports the comparison results of different thresholds on CIFAR-100. We can observe that the performance of the proposed method is not sensitive to $\epsilon_1$ and $\epsilon_2$, which can achieve superior performance within a certain range of thresholds. Especially, all results of the proposed method are better than the baselines under the joint *(x,y)-noise* in Table 3. In summary, our method indeed shows the effectiveness and practicability of dealing with noisy data, which does not rely on time-consuming hyper-parameters tuning.

Table 3: The comparison of validation accuracy under different $\epsilon_1$ and $\epsilon_2$ on CIFAR-100 with different levels of *(x,y)-noise*.

| $\epsilon_1$ ($10^{-2}$) | | 1.5 | 2.0 | 2.5 | 3.0 | 3.5 | $\epsilon_2$ (%) | 2.0 | 3.0 | 4.0 | 5.0 | 6.0 |
|---|---|---|---|---|---|---|---|---|---|---|---|---|
| $0.4y+0.3x$ | Best | 66.01 | 66.61 | 66.57 | 66.34 | **66.58** | Best | 64.92 | 65.81 | 66.02 | **66.72** | 66.11 |
| | Last | 65.63 | 66.05 | 66.04 | **66.10** | 66.01 | Last | 64.43 | 64.52 | 65.59 | **66.09** | 66.04 |
| $0.2y+0.3x$ | Best | 71.92 | **72.78** | 72.61 | 72.69 | 72.71 | Best | 71.93 | 72.51 | 72.76 | **72.93** | 72.88 |
| | Last | 71.22 | **72.48** | 72.24 | 72.45 | 72.44 | Last | 71.85 | 72.33 | 72.24 | **72.59** | 72.49 |
| $0.6y$ | Best | 57.85 | **59.65** | 59.52 | 58.59 | 57.94 | Best | 58.20 | 58.92 | 58.99 | 59.07 | **59.61** |
| | Last | 55.96 | **55.53** | 55.26 | 55.06 | 55.03 | Last | 57.01 | 56.63 | 56.17 | **56.44** | 55.21 |
| $0.4y$ | Best | 70.63 | **70.77** | 70.28 | 70.11 | 69.88 | Best | 69.82 | 69.98 | **70.81** | 70.62 | 70.59 |
| | Last | 68.27 | **68.83** | 68.19 | 67.63 | 67.46 | Last | 67.89 | 68.07 | 68.94 | 68.71 | **68.72** |

**Effects of uncertainty penalty in the proposed learning objective.** To verify the effectiveness of uncertainty penalty in Eqn. (10), we report the performance of the proposed method without uncertainty penalty and the gap with "Proposed-LM" in Table 4. We observe that the validation

accuracy is lower than the complete workflow on both *y-noise* and *(x,y)-noise*, which clarify that the effectiveness of the uncertainty penalty strategy.

Table 4: The best accuracy on CIFAR-100 and TinyImageNet with *(x,y)*-noise.

| Noise rate | $0.4y$ | $0.6y$ | $0.2y+0.3x$ | $0.4y+0.3x$ |
|---|---|---|---|---|
| | CIFAR-100 / TinyImageNet | | | |
| Best Acc | 67.92/54.23 | 57.33/41.52 | 70.81/55.34 | 63.02/49.68 |
| Gaps | 2.85/1.98 | 2.17/3.13 | 1.97/3.41 | 3.59/2.67 |

## 6 Conclusions and limitations

### 6.1 Conclusions

This work first introduces the more challenging and closer to real-world noise setting and then performs a systematical investigation on using uncertainty-based models under dual noises (i.e., the joint *(x,y)-noise*). We find that merely employing an uncertainty-based model is not enough and furthermore propose a novel workflow for the learning of uncertainty-based deep models. Concretely, we present the efficient and practical confidence-based sample filter to distinguish noisy data from clean data progressively. After doing so, we propose to penalize the model uncertainty of noisy data without reliance on the misleading supervisory information. Empirically, the proposed method significantly outperforms the competitive baselines on CIFAR-100 and TinyImageNet with synthetic *(x,y)-noise* and the real-world noisy dataset. We further evaluate the robustness of hyper-parameters in our method, which shows that the proposed method is not sensitive to crucial hyper-parameters. In the future, this work may promote more approaches to deal with dual noises in more tasks.

### 6.2 Potential Limitations

Though the proposed method shows superior performance, there are also some potential limitations. First, one limitation of this work is that we have not separately handled the two kinds of *x-noise* for better noise detection and model training, so designing a more fine-grained approach might be helpful for model learning, and it deserves future investigation. Second, deep ensemble is usually computationally expensive especially when the model and data complexity is high. To address the limitation, we can use computationally cheap deep ensemble but with comparable performance, such as BatchEnsemble [47] and Hyperparameter ensembles [48], whose computational and memory costs are significantly lower than typical ensembles.

## Acknowledgement

This work was supported by National Key Research and Development Project of China (No. 2021ZD0110502), NSFC Projects (Nos. 62061136001, 62076145, 62076147, U19B2034, U1811461, U19A2081, 61972224), Beijing NSF Project (No. JQ19016), BNRist (BNR2022RC01006), Tsinghua Institute for Guo Qiang, and the High Performance Computing Center, Tsinghua University. J.Z is also supported by the XPlorer Prize.

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
