

(a) Histogram of **L-Con** at 40th epoch.

(b) Histogram of **L-Con** at 100th epoch.

Figure 4: **L-Con** distributions at training phase on CIFAR-100 with 40% *y-noise*.

# A    The performance of different uncertainty-based models under dual noises

Table 5: The comparison of validation accuracy on CIFAR-100 with the joint *(x,y)-noise*. "$0.2y+0.3x$" represents the dataset with 20% *y-noise* and 30% *x-noise* simultaneously.

| Methods/Val. | | $0.2y+0.3x$. | $0.4y+0.3x$ |
|---|---|---|---|
| Single-CE | Best | 57.84 | 47.76 |
| | Last | 57.39 | 41.39 |
| BNNs | Best | 62.25 | 51.20 |
| | Last | 61.96 | 50.68 |
| SNGP | Best | 59.92 | 49.27 |
| | Last | 59.22 | 49.09 |
| DE-CE | Best | **66.50** | **54.90** |
| | Last | **65.24** | **53.91** |

We also explore more uncertainty-based models in addition to deep ensemble to fit the noisy data with dual noises, e.g., Bayesian neural networks (BNNs) with mean-field variational inference (MFVI) and Spectral-normalized Neural Gaussian Process (SNGP) [44]. We do not consider the uncertainty-based models (e.g., Monte Carlo (MC) dropout [15], DUQ [29] and Prior Network [33]) that can not explicitly model uncertainty at the training phase in our experiments because the lack of uncertainty can not alleviate over-fitting during the training time. Table 5 presents the classification accuracy of uncertainty-based models and deterministic DNNs on CIFAR-100 with dual noises. We can see that uncertainty-based models can better alleviate over-fitting than deterministic DNNs. Especially, deep ensemble used in this paper can achieve the best performance compared to BNNs and SNGP, so we opt to place our workflow on the well-evaluated deep ensemble to establish a strong learning approach under dual noises.

Table 6: The AUROC scores of detecting noisy samples with different levels of *x-noise* and *y-noise*.

| Noise rate | $0.3x$ | $0.4x$ | $0.3y$ | $0.4y$ |
|---|---|---|---|---|
| AUROC | 89.10% | 88.13% | 95.21% | 94.02% |

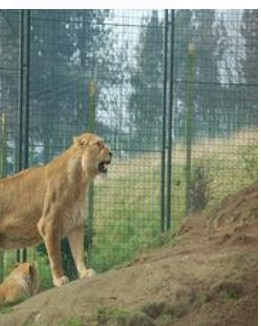 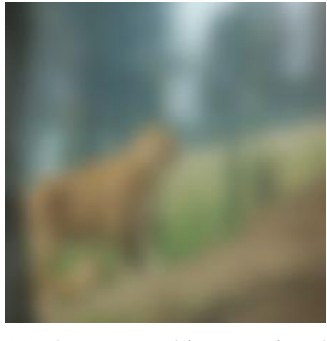 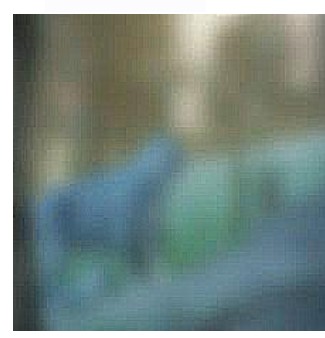 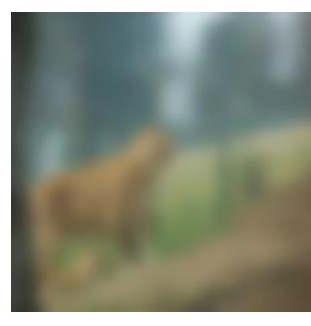

(a) The corrupted image using the Gaussian blur.

(b) The restored image using the mean filter in *OpenCV*.

(c) The restored image using the latest denoising technique based on NNs [4].

Figure 5: Blurred and restored images using different image denoising methods.

## B  Experiment details

### B.1  Preprocessing

All images are normalized and augmented by random horizontal flipping. For CIFAR-100, we use the standard $32 \times 32$ random cropping after zero-padding with 4 pixels on each side. For TinyImageNet, we use randomly crop a patch of size $56 \times 56$ from each image. For WebVision, we first resize each image to make the size as 320. Then we use the standard data augmentation, randomly crop a patch of size $299 \times 299$ from each image, and apply horizontal random flipping.

### B.2  Optimizer and hyper-parameters

SGD with momentum (0.9) and weight decay $3 \times 10^{-4}$ is used in all experiments. For the setting of the thresholds $\epsilon_1$ and $\epsilon_2$, we recommend performing a grid search for $\epsilon_1 \in \{0.015, 0.020, 0.025, 0.030, 0.035\}$ and $\epsilon_2 \in \{1\%, 2\%, 3\%, 4\%, 5\%, 6\%\}$ to achieve the better performance. In the experiments, we set $\epsilon_1 = 0.020$ and $\epsilon_2 = 5\%$ for CIFAR-100 and TinyImageNet.

### B.3  Network

Five networks with ResNet-18 are trained from scratch using PyTorch 1.9.0. for all experiments. Default PyTorch initialization is used on all layers. It is noteworthy that we need to use the small convolution with $3 \times 3$ kernel in the downsampling layer for CIFAR-100 and TinyImageNet.

### B.4  Warm-up

The model warm-up can help better separate noisy data and clean data. We start training the model with high learning rates and standard cross-entropy loss in experiments. Specifically, our method uses the learning rate of 0.2 for the first 35 epochs for CIFAR-100 and TinyImageNet. For WebVision, we use the learning rate of 0.2 for the first 40 epochs.

### B.5  Training schedule

For CIFAR-100 and TinyImageNet, training for 250 epochs in total, and we reduce the initial learning rate (0.2) by a factor of 2.5 after 35, 80, 120, 150 and 180 epochs. For WebVision, we train the model for 130 epochs and reduce the initial learning rate (0.1) by a factor of 10 after 80 and 105 epochs.

## C  Performance under synthetic y-noise

We report the results of all methods only under label noise in order to directly compare the proposed method with previous works focusing on this setting. Table 7 presents the results on CIFAR-100 and TinyImageNet with different levels of *y-noise*, which shows that the proposed method significantly

Table 7: The comparison of validation accuracy on CIFAR-100 and TinyImageNet with *y-noise*.

| Alg./Noise rate | | 0.0 | 0.1 | 0.2 | 0.4 | 0.6 |
|---|---|---|---|---|---|---|
| | | CIFAR-100 / TinyImageNet | | | | |
| DE-CE | Best | 79.13/63.62 | 75.30/60.03 | 71.19/55.82 | 60.65/47.89 | 51.26/39.14 |
| | Last | 77.01/61.28 | 75.16/58.06 | 70.84/53.02 | 59.16/41.00 | 42.74/32.36 |
| DYR [1] | Best | 78.64/**65.14** | 73.76/60.04 | 68.47/56.33 | 58.43/47.85 | 46.02/37.19 |
| | Last | 78.02/63.97 | 73.13/58.13 | 67.31/54.28 | 57.06/45.72 | 44.91/35.86 |
| M-DYR [1] | Best | 75.38/62.32 | 75.43/60.40 | 75.18/**59.59** | 69.43/54.59 | 59.48/42.06 |
| | Last | 74.91/61.04 | 75.12/59.11 | 74.69/**58.25** | 68.73/52.77 | 56.07/41.26 |
| CORES$^2$ [8] | Best | 76.76/59.74 | 71.79/57.15 | 67.42/54.53 | 55.18/46.95 | 42.97/37.17 |
| | Last | 76.22/59.14 | 71.03/57.00 | 66.62/53.26 | 54.31/45.39 | 41.89/36.02 |
| Proposed-L | Best | 80.44/64.07 | **79.45**/60.25 | **75.76**/58.96 | 69.44/59.02 | 58.76/43.00 |
| | Last | 79.03/63.21 | 77.15/59.31 | 74.89/58.87 | 66.04/**55.98** | 57.02/41.54 |
| Proposed-LM | Best | **80.98**/64.58 | 79.32/**60.73** | 75.20/58.65 | **70.77/56.21** | **59.52/44.65** |
| | Last | **79.71/64.15** | **78.81/59.95** | **75.15**/58.01 | **68.83**/55.84 | **56.26/44.10** |

outperforms competitive methods for noisy labels (i.e., DYR, M-DYR, and CORES$^2$). The results verify that our method is also suitable for scenarios that only involve *y-noise*. In particular, the proposed methods (i.e., no matter Proposed-L or Proposed-LM) also exhibit superior performance on clean data(i.e., 0% noise). The result illustrates the proposed sample filtering and learning strategy is robust and can not bias the learning of the model on overall clean datasets, which is not enjoyed by other methods.

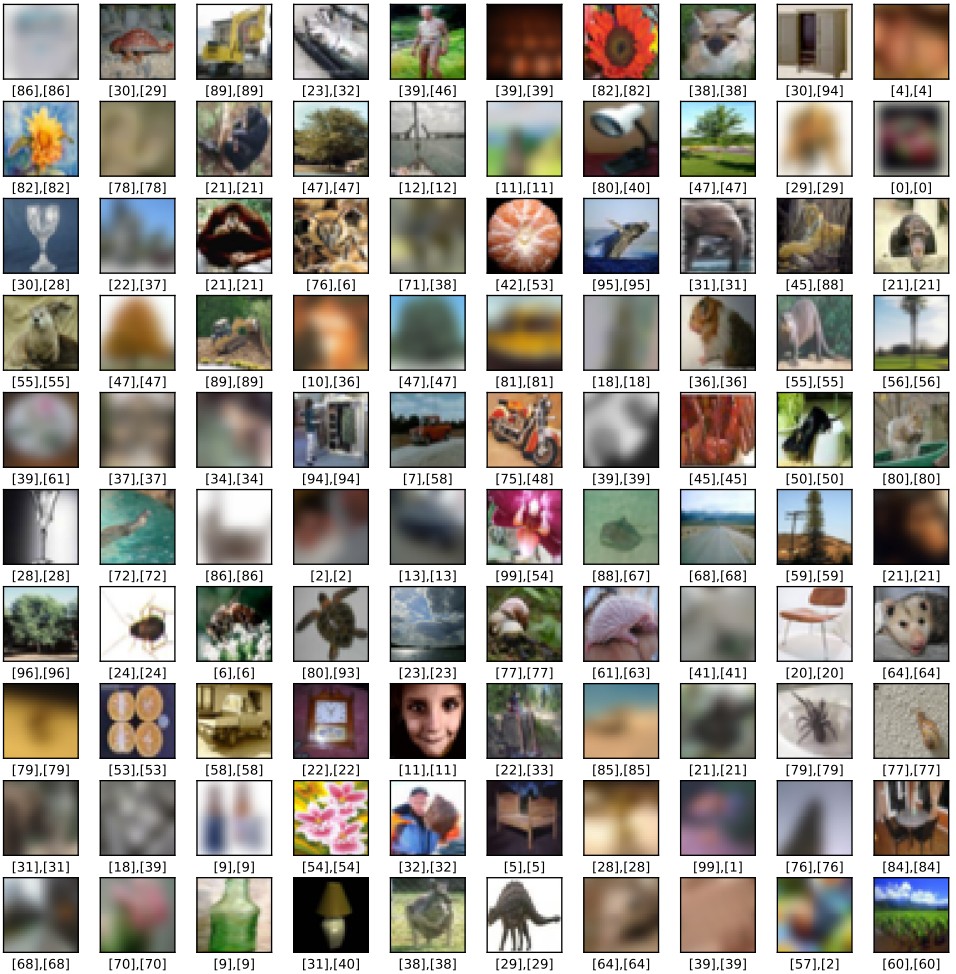

Figure 6: Some random images of CIFAR-100 with dual noises. The two numbers below each image represent the actual label id and the correct label id respectively. If two ids are not identical, it indicates an image with label noise.

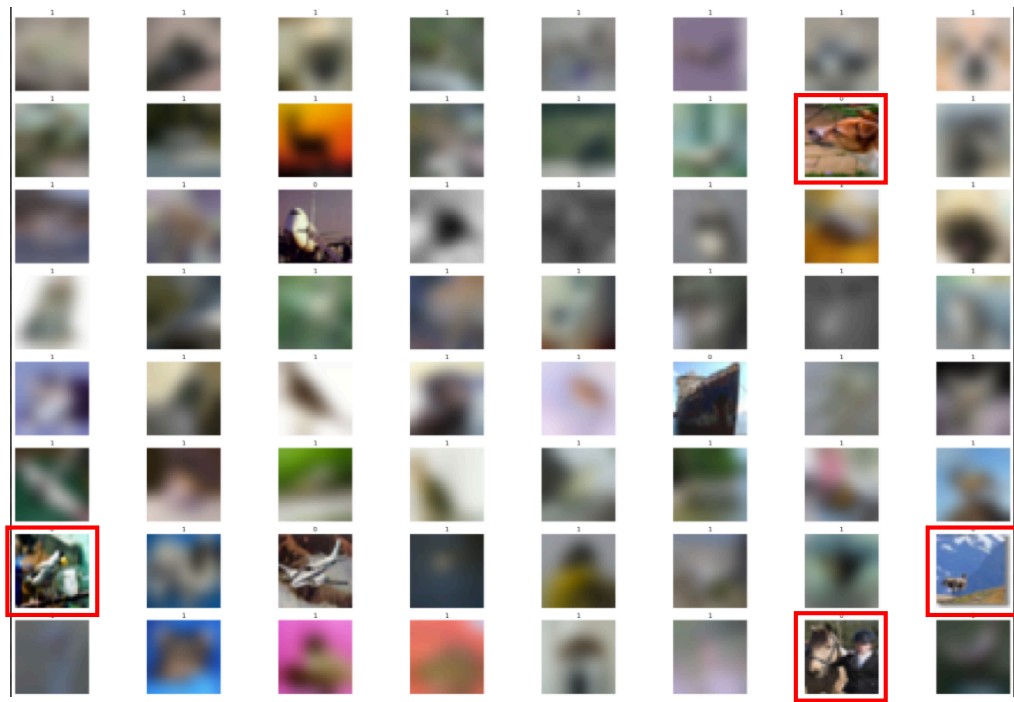

Figure 7: There are some low **M-Cons** corresponding images. Images with red boxes represent hard or dirty samples (e.g., some images contain multiple objects or some images contain background noise).