# OpenReview forum: "Confidence-based Reliable Learning under Dual Noises"
_NeurIPS.cc/2022/Conference — NeurIPS 2022 Accept_

### Official Review · Reviewer_uLBd · 2022-07-07

**Rating:** 5
**Confidence:** 4
**Soundness:** 3 good
**Presentation:** 3 good
**Contribution:** 3 good

**Summary:**

This work explores a novel learning task of supervised learning with both image noise and label noise. The focus on the learning under dual noises seems to assume a maximum of one type of noise in each data point (unclear to me). To address the proposed task, a unified framework with a confidence-based sample filter is proposed, where model uncertainty is penalized instead to avoid over-fitting with false information.

**Questions:**

- Please clarify if a given image label will only have x-noise or y-noise only? or both types of noise can be applied

- For the section in line 156-166, please discuss the challenge of detecting x-noise only in the easy and challenging datasets. I have concerns about this as on a hard dataset, the data uncertainty could come from either noise or the underlying complexity/bias of the visual data.

- In line 188, it states that the dirty observation is unhelpful for model training. I would like the author to elaborate if x-noise I and x-noise II would have a similar impact on the model learning, and shall both types of noise be handled differently.

- Please also discuss if the parameters (e.g., epsilon_1 and epsilon_2) are dataset-dependent. It is unclear if tuning is required for different datasets.

- I suggest naming the WebVision as WebVision-50 in the paper as readers may think it is validated on the full dataset.

- In Fig 3, is the Best performance or Last reported results.

- For Table 1, please also show result with clean dataset (no noise augmented). This will give the reader a clear view of the challenge in present of noise.

**Ethics Review Area:**

["I don’t know"]

**Limitations:**

The paper discusses the technical limitation but not the societal impact. However, I don't see any potential societal impact that is unique in this work.

**Strengths And Weaknesses:**

+ This work advances the field by considering a new task that simultaneously handles both image noise (x-noise) and label noise (y-noise) presented in the training dataset. The considered problem is highly realistic and would be very useful for learning with real-world data.
+ This work proposes a confidence-based method to detect images that may have noise. Then, dedicated thresholds are used to determine if a given image is noisy or not.
+ A learning objective is derived to estimate model uncertainty, which is subsequently used as a learning objective to improve the discriminative and robustness of the trained model.

- One concern of the weakness is that the filtering is a binary process (eqn 6 & 7). As the experiment is conducted on CIFAR-100 and Tiny ImageNet, it is unclear if such a naive approach is reliable for more challenging real-world datasets (e.g., ImageNet).

---

> ### Author Response · Authors · 2022-08-02
> **Author Response**
>
> We appreciate the acknowledgment of the practicability of this work. We address the detailed comments below. We hope that you may find our response satisfactory and raise your score accordingly.
>
> **Q1: It is unclear if such a naive approach is reliable for more challenging real-world datasets (e.g., ImageNet):** Although ImageNet has more classes and images, the effect of dual noise on training data is essentially similar to CIFAR and TinyImageNet. Hence, intuitively, the label confidence (**L-Con**) and max confidence (**M-Con**) can still indicate noisy labels and noisy images, respectively. This is also verified by the previous work [*1]. Moreover, TinyImageNet is a subset of ImageNet, and its experiment results can imply the efficacy of the proposed method on ImageNet to some extent.
>
> **Q2: Does a given image-label have only x-noise or y-noise?** No, it is assumed that these two types of noise are applied independently to (image, label) pairs, which are closer to the real world. Specifically, some training samples may have both kinds of noise, and some training samples may have only one of them.
>
> **Q3: The challenge of detecting x-noise:** Thanks for the comment. We agree that the data uncertainty could come from either noise or the underlying complexity of visual data. As evidenced by some closely related works [*2, *3], detecting the data with underlying complexity/bias for a particular treatment can improve the model's predictive performance or training efficiency. In fact, the **M-Con** filter in our method is able to detect the data with underlying complexity/bias: as shown in Fig. 7 of the appendix, some samples with low **M-con** correspond to the data with underlying complexity or bias (i.e., hard or dirty samples), and filtering out these samples does not compromise predictive accuracy. Concretely, as evidenced by the validation results in Table 1, we can observe that the proposed method also exhibits superior performance on clean data (i.e., 0\% noise). We will add these discussions to the revision.
>
> **Q4: Do x-noise I and x-noise II similarly impact the model learning?** We empirically observe that x-noise I is more detrimental to model learning than x-noise II. This is because that the image with x-noise II still contain valuable semantic information but severely corrupted images (e.g., *Gaussian Blur*, *Fog*, and *Contrast*) do not. To maintain the unity and simplicity of the proposed method, we deal with the two types of x-noise in the same way with the **M-con** filter. Separately coping with them is viable, and we can draw insights from recent works like [*4] and [*5] to dedicatedly handle x-noise II. We agree that more fine-grained treatment of the two kinds of x-noise may be helpful for model learning, and it deserves future investigation.
>
> **Q5: About the robustness of parameters (e.g., $\epsilon_1$ and $\epsilon_2$) :** $\epsilon_1$ and $\epsilon_2$ are dataset-independent, we use the same hyper-parameters (i.e., $\epsilon_1=0.02$ and $\epsilon_2=5\%$) for all datasets in our experiments. Specifically, the experiments on various $\epsilon_1$ and $\epsilon_2$ demonstrate the robustness and stability of our method against the hyper-parameters in Sec. 5.3. We will make it clearer in the next version.
>
> **Q6: The name of the WebVision:** Thanks for your suggestion, and we will make it more precise in the final version.
>
> **Q7: The results of Fig. 3:** Fig. 3 reports the comparison on the best validation accuracy, and we will clarify it in the final version.
>
> **Q8: About the result with clean dataset:** Thanks. The results on clean datasets have been reported in Table 6 in Appendix C, and we will also add them to Table 1 to give the reader a clear view of the challenge in the presence of noise.
>
> **References:**
>
> [*1] Dan Hendrycks et al. Deep Anomaly Detection with Outlier Exposure. ICLR2019.
>
> [*2] Jie Chang et al. Data Uncertainty Learning in Face Recognition. CVPR2020.
>
> [*3] Sören Mindermann et al. Prioritized Training on Points that are learnable, Worth Learning, and Not Yet Learnt. ICML2022.
>
> [*4] Bohan Zhuang et al. Attend in groups: A weakly-supervised deep learning framework for learning from web data. CVPR2017.
>
> [*5] Yi Tu et al. Learning from web data with self-organizing memory module. CVPR2020.

---

> > ### Comment · Reviewer_uLBd · 2022-08-10
> > **Response**
> >
> > Thanks for the detailed response. It does address most of my concerns.
> >
> > I just want to point out that despite the experiment on Tiny ImageNet, it is still unknown about the impact of dual noises in the real-world scenario, where images may have higher quality and exhibit higher diversity in the visual context. This will be a good and important direction for the research community.

---

### Official Review · Reviewer_Lg6P · 2022-07-11

**Rating:** 4
**Confidence:** 3
**Soundness:** 2 fair
**Presentation:** 3 good
**Contribution:** 2 fair

**Summary:**

This paper proposes a unified framework to address the noise in the input images and annotated labels. To be more specific, the proposed method uses confidence scores to distinguish if the labels or the input images are noisy or not.

**Questions:**

1. The authors might need to carefully design how to set confidence scores of noisy label and input images.
2. The authors might need to explain more on the selections of epsilons, which are a very important part of the proposed method.


**Limitations:**

The authors have addressed the potential societal impact.

**Strengths And Weaknesses:**

Strength:
1. The paper is well written and clearly organized, and the paper is easy to follow.
2. The proposed method is easy to implement.
3. The proposed method is able to address the noise in the input images and labels at the same time.

Weaknesses:
1. The proposed method seems trivial. In the equation 6 and 7, the weights are only determined by using two thresholds, which are fine tuned by hand. This makes the proposed method trivial and incremental.
2. Also the reviewer has some concerns about the equation 6 and 7. The hyper-parameters epsilon 1 and epsilon 2 are tuned by hand and in the appendix B.2, the authors claimed that they are set to 0.02 and 5% respectively. The reviewer is curious about the selection of these two hyper-parameters. Do these parameters require heavy tunes or is it complicated to tune epsilon 1 and 2? If not, why do the authors conduct experiments on epsilon 1 from 0.015 to 0.035, and epsilon 2 from 1% to 6%? Further explanations might be needed on these parameters.
3. To the best of the reviewer's knowledge, a large-scale noisy dataset is not included in the paper - ImageNet-C, which contains different types of noises. The reviewer is wondering why the authors ignore this benchmark.

---

> ### Author Response · Authors · 2022-08-02
> **Author Response**
>
> We thank you for the valuable comments and answer the specific questions below. We hope that you may find our response satisfactory and raise your score accordingly.
>
> **Q1: The proposed method seems trivial given manually tuned thresholds:** We do not agree with this comment with our highest respect. In fact, to the best of our knowledge, this paper provides the first unified solution to tackle the dual noise on both the observations and labels, as agreed by Reviewers 4VmN and uLBd. Reviewers 4VmN and uLBd also found the `dual noise' problem realistic, interesting, and important. Thus, the main technical novelty of this paper lies in that the developed confidence-based filter can effectively detect the two kinds of noisy data while being easy to implement. Besides, as evidenced by the validation results in Table 3 (see the answer to Q2), our whole methodology is robust against the confidence thresholds, so the cost for hyper-parameter tuning can be moderate in practice. Finally, the uncertainty penalty strategy is also our valuable technological innovation.
>
> **Q2: The hyper-parameters $\epsilon_1$ and $\epsilon_2$ :** Thanks for the comments. The direct answer is that the proposed method does not rely on time-consuming hyper-parameters tuning. The validation results corresponding to various $\epsilon_1$ and $\epsilon_2$ in various scenarios are reported in Table 3. As shown, the final performance of the trained models does not change drastically within a range of selections of $\epsilon_1$ and $\epsilon_2$. Especially, all the results of our method are better than the baselines under the joint (x,y)-noise (i.e., ''0.2y+0.3x'' and ''0.4y+0.3x'' in Table 3). Thanks to such a nice property, we use the same hyper-parameters (i.e., $\epsilon_1=0.02$ and $\epsilon_2=5\%$) to conduct experiments across multiple datasets. To summarize, the experiments on various $\epsilon_1$ and $\epsilon_2$ demonstrate the robustness and stability of our method against the hyper-parameters. We'll make clear this point in the revision.
>
> **Q3: A noisy large-scale dataset is not included in the paper--ImageNet-C:** Thanks for the suggestion. We clarify that (1) the current experiments are thorough, as acknowledged by Reviewer 4VmN; (2) as some corruptions in ImageNet-C are popular data augmentation that will facilitate the training of the classifier rather than threaten it, we apply only the ''detrimental'' *Gaussian Blur*, *Fog*, and *Contrast* to natural images from CIFAR and TinyImageNet to form challenging testbeds. The existing results have already provided strong support for the claims made in the paper and indicated the superiority of the proposed method over typical baselines for handling dual noise. Moreover, the results on TinyImageNet can imply the efficacy of the proposed method on ImageNet to some extent. Anyway, we'll try to add the direct results on ImageNet-C in the final version.

---

> ### Author Response · Authors · 2022-08-09
> **Looking forward to your feedback**
>
> Dear Reviewer Lg6P ,
>
> Thanks again for your valuable comments ! We have responded to your initial comments. We are looking forward to your feedback and will be happy to answer any further questions you may have in the author-reviewer discussion period. If all your concerns have been resolved, it is much appreciated if you may raise the rating of our work.
>
> Thank you,
>
> authors

---

### Official Review · Reviewer_4VmN · 2022-07-11

**Rating:** 7
**Confidence:** 3
**Soundness:** 3 good
**Presentation:** 3 good
**Contribution:** 4 excellent

**Summary:**

This paper addresses the problem of joint image and label noise. It proposes a unified framework for reliable learning to deal with the joint noise problem. It first filters out noisy data and then utilizes deep ensemble to decrease the model uncertainty of noisy data. The experiments reveal that the proposed model achieves superior performance on the dataset with real-world noise.

**Questions:**

1. The relationship "noisy image v.s. value of w" and "noisy label v.s. value of w" should be demonstrated or visualized to validate the effectiveness of the filter.
2. The authors should discuss the limitations of the work.

**Limitations:**

The limitations and potential negative societal impact are not addressed.

**Strengths And Weaknesses:**

1. This paper addresses a very important and interesting problem, which is noise exists in both images and labels. The paper is well written and easy to follow.
2. This paper proposes the first unified framework for joint noise by distinguishing noisy data from clean data based on the practical confidence sample filter. It is promising the intuitive to penalize the model uncertainty of noisy data.
3. There are plenty of experiments in the paper to clarify the effectiveness of the model design such as the uncertainty penalty strategy. The proposed method significantly outperforms state-of-the-art methods on the real-world noisy dataset.

---

> ### Author Response · Authors · 2022-08-02
> **Author Response**
>
> Thank you for appreciating our new contributions as well as providing valuable comments. We address the detailed concerns below.
>
> **Q1: The relationship ''noisy image v.s. value of w'' and ''noisy label v.s. value of w'' should be demonstrated:** We appreciate the constructive suggestion. In fact, we have visualized distributions of **L-Con** (corresponding to ``noisy label v.s. value of w'') in Fig. 4 of the Appendix. Besides, Table 3 presents how the threshold for estimating w affects the validation accuracy, which can implicitly reflect the effectiveness of the filters.
>
> Furthermore, we perform new experiments to demonstrate the efficacy of the filters for selecting noisy images and noisy labels. Concretely, we regard the detection of noisy data as a binary classification problem and use the Area Under the Receiver Operating Characteristic curve (AUROC)  to indicate the effectiveness of our filter. We observe that, at the 60th epoch of the model training under 40\% y-noise and 30\% x-noise on CIFAR-100, the AUROC of the **L-Con** filter is 96\%, and that of the **M-Con** filter is 89\%. We will add more comprehensive results regarding this in the revision.
>
> **Q2: The limitations of the work:** Thanks for the kind suggestion. We are sorry that we have not adequately discussed the limitations of this work due to the page limit. As mentioned in Sec 4.1, one limitation of this work is that we have not separately handled the two kinds of x-noise for better noise detection and model training. As a fundamental research paper, this work has no significant negative impact on society. We will add more discussions to the revision.

---

### Author Response · Authors · 2022-08-08
**Looking forward to further feedback.**

Dear Reviewers,

Thank you for your thorough reviews. We are looking forward to your reply. If you have any further concerns or requests, we can have a chance to address them in the author-reviewer discussion period. If all your concerns have been resolved, it is much appreciated if you may raise the rating of our work.

Thank you,

authors

---

### Meta-Review · Area_Chair_rB6M · 2022-08-31

**Recommendation:** Accept
**Confidence:** Certain

**Metareview:**

Novel work that addresses the somewhat realistic setting of noise in input (images) and annotated labels. The method uses confidence scores to decide whether the labels or the images are noisy. Reviewers agree that the work is clear and easy to follow + easy to implement. I disagree with the reviewer that the proposed method is trivial -- fundamentally, showing that these simple thresholds actually help in a variety of settings is in fact a useful contribution to the community. I believe the simplicity of the proposed approach will make it more likely that this work is adopted or studied in the community, so I recommend acceptance.

**Award:**

No

---

### Decision · Program_Chairs · 2022-09-14

Accept